# A Comprehensive Phytochemical Analysis of *Sideritis scardica* Infusion Using Orbitrap UHPLC-HRMS

**DOI:** 10.3390/molecules29010204

**Published:** 2023-12-29

**Authors:** Dimitrina Zheleva-Dimitrova, Yulian Voynikov, Reneta Gevrenova, Vessela Balabanova

**Affiliations:** 1Department of Pharmacognosy, Faculty of Pharmacy, Medical University of Sofia, 1000 Sofia, Bulgaria; rgevrenova@pharmfac.mu-sofia.bg (R.G.); vbalabanova@pharmfac.mu-sofia.bg (V.B.); 2Department of Chemistry, Faculty of Pharmacy, Medical University of Sofia, 1000 Sofia, Bulgaria; y_voynikov@pharmfac.mu-sofia.bg

**Keywords:** *Sideritis scardica* Griseb, UHPLC–HRMS, phytochemical analysis

## Abstract

*Sideritis scardica* Griseb, also known as “mountain tea” and “Olympus tea” (Lamiaceae family) is an endemic plant from the mountainous regions of the Balkan Peninsula. In this study, we focused on an in-depth phytochemical analysis of *S. scardica* infusion using ultra-high-performance liquid chromatography hyphenated with high-resolution mass spectrometry (UHPLC–HRMS). Quantitative determination of the main secondary metabolites was carried out by UHPLC–HRMS analyses using the external standard method. The results revealed more than 100 metabolites, including five sugar acids and saccharides, 21 carboxylic, hydroxybenzoic, hydroxycinnamic acids, and derivatives, 15 acylquinic acids, 10 phenylpropanoid glycosides, four iridoid glycosides, 28 flavonoids, seven fatty acids, and four organosulfur compounds. Furthermore, a dereplication and fragmentation patterns of five caffeic acids oligomers and four acylhexaric acids was performed for the first time in *S. scardica*. Regarding the quantitative analysis, the phenylethanoid verbascoside (**53**) (151.54 ± 10.86 mg/g lyophilized infusion, li), the glycosides of isoscutellarein (**78**) (151.70 ± 14.78 mg/g li), methylisoscutelarein (**82**) (107.4 ± 9.07 mg/g li), and hypolaetin (**79**) (78.33 ± 3.29 mg/g li), as well as caffeic acid (**20**) (87.25 ± 6.54 mg/g li), were found to be the major compounds in *S. scardica* infusion. The performed state-of-the-art phytochemical analysis of *S. scardica* provides additional knowledge for the chemical constituents and usage of this valuable medicinal plant.

## 1. Introduction

*Sideritis scardica* Griseb (Lamiaceae family) is an endemic plant of the mountainous regions of the Balkan Peninsula [1,2]. It is often referred to as “mountain tea”, “ironwort”, “Olympus tea”, and “Pirin tea” [3]. Mountain tea is a perennial herbaceous plant with a well-developed root system, the stem is 15–40 cm and woody at the base, the leaves are opposite with gray hairs, the flowers are clustered in a dense spike, the middle bracts are 12–20 mm long, i.e., longer than the flowers, the corolla is lemon yellow with glandules, and the calyx is tubular-campanulate [2,3]. Usually, *Sideritis* plants are applied in traditional medicine, mostly as an aromatic herbal tea [4,5,6,7]. The tea is made from the aerial parts of the plant by infusion or decoction [8]. Historically, *S. scardica* has been used to treat inflammation, common colds, asthma, bronchitis, and gastrointestinal disorders. It is supposed to relieve pain, including rheumatic pain, as well as reducing stress and anxiety. The plant name comes from the Greek word “sideros”, meaning “iron”, as it was used in ancient times to heal wounds from iron weapons [4]. Regular consumption of mountain tea by rats has been shown to lead to weight loss and prevent insulin resistance by lowering blood glucose and triglyceride levels and increasing liver glycogen content [8]. Additionally, antioxidant properties and positive effects on memory and cognitive abilities have also been observed [7,9,10]. *Sideritis* species have also been used topically on the skin and as an antiseptic solution to sooth the pain of tooth extraction [3].

The traditional medicinal usage of the species is based on the phytochemical constituents, including phenolic acids (chlorogenic acid, 3-caffeoylquinic acid, feruloylquinic acid, and others), flavonoids and their derivatives (hypolaetin, isoscutellarein, and others), phenylethanoid glycosides (lavandulifolioside, verbascoside, echinacoside, allysonoside, and others), and terpenoids (mostly iridoid glycosides) [6,8,11,12]. These chemical compounds have been explored in phytochemical studies, operating with various extraction techniques such as hydrodistillation and solvent and supercritical extractions [8,12,13,14,15]. Precisely, the most abundant secondary metabolites of *S. scardica* water extracts (i.e., when making infusion or decoction) are flavonoids, hydoxycinnamic acid derivatives, and phenylethanoid glycosides [7,14,16,17]. Identification of closely related species of the genus *Sideritis* is based on the dominant 5-caffeoylquinic acid, lavandulifolioside, verbascoside, isoscutellarein 7-O-allosyl(1→2)glucoside, hypolaetin 7-O-[6‴O-acetyl]-allosyl(1→2)glucoside, isoscutellarein 7-O-[6‴-O-acetyl]-allosyl(1→2) glucoside, 3′-O-methylhypolaetin 7-O-[6‴-O-acetyl]-allosyl(1→2)glucoside, and 4′-O-methylhypolaetin 7-O-[6‴-O-acetyl]-allosyl-(1→2)-[6″-O-acetyl]-glucoside. Thus, these compounds have been used as chemotaxonomical markers [6]. 

Based on the literature available on *Sideritis scardica*, there is no detailed metabolite profiling of the species, something which seems important in light of its health benefits. An in-depth UHPLC–HRMS analysis of the main metabolites of *S. scardica*, together with quantitative determination, was conducted. More than 100 secondary metabolites were identified/tentatively elucidated in a lyophilized infusion of mountain tea. The performed phytochemical analysis of *S. scardica* will provide additional knowledge of the chemical constituents and usage of this valuable medicinal plant for the future.

## 2. Results and Discussion

### 2.1. Metabolite Profiling of S. scardica Lyophilized Infusion

Herein, an in-depth UHPLC–HRMS analysis of *S. scardica* infusion was conducted by allowing the dereplication/annotation of 103 metabolites, including five sugar acids and saccharides, 21 carboxylic, hydroxybenzoic, hydroxycinnamic acids, and derivatives, five caffeic acids oligomers, 15 acylquinic acids, four acylhexaric acids, 10 phenylpropanoid glycosides, four iridoid glycosides, 28 flavonoids, seven fatty acids, and four organosulfur compounds (Table 1). This study allowed the identification of caffeic acids oligomers, acylquinic, acylhexaric acids, and flavonoids not previously reported in the taxon. The total ion chromatogram (TIC) in negative ion mode of the studied extract is presented in Figure 1.

#### 2.1.1. Sugar Acids and Saccharides 

Compound **1** ([M−H]^−^ at *m*/*z* 165.041) gave fragment ions at *m*/*z* 147.02 [M−H−H_2_O]^−^, 129.018 [M−H−2H_2_O]^−^, 111.01 [M−H−3H_2_O]^−^, and 101.023 [M−H−3H_2_O−CO]^−^, as well as ions corresponding to the loss of 60 Da [M−H−C_2_H_4_O_2_]^−^ and 90 Da [M−H−C_3_H_6_O_3_]^−^, respectively. Thus, compound **1** was annotated as xylonic acid. Analogously, **4** was related to pentose with a base peak at *m*/*z* 75.00 (C_2_H_2_O_3_). Moreover, **2** (with an additional CH_2_ group compared to **4**) and **3** (with additional CH_2_O) were identified as hexose and gluconic acid, respectively (Table 1). The identity of asystoside (**5**) was suggested by the transitions 583.261→421.209→289.166→161.445, resulting from the losses of hexosyl (162.053 Da), pentosyl (132.043 Da), and oct-1-en-3-ol units (C_8_H_16_O, 128.122 Da) (Table 1). All above-mentioned compounds were previously identified in the species [12].

#### 2.1.2. Carboxylic, Hydroxybenzoic, Hydroxycinnamic Acids and Their Derivatives

Eight hydroxybenzoic acids (**9**, **12**, **15**, **16**, **18**, **19**, **21**, and **24**), four hydroxycinnamic acids (**17**, **20**, **25**, and **26**) and their glycosides (**10**, **11**, **13**, **14**, **22**, and **23**), together with quinic (**6**), oxaloglutaric (**7**), and citric acid (**8**), were identified based on the comparison with reference standards and literature data in the assayed extract (Table 1) [12,17]. Compound **8** ([M−H]^−^ at *m*/*z* 191.018) showed fragment ions at *m*/*z* 173.008 [M−H−H_2_O]^−^, 147.028 [M−H−CO_2_]^−^, a base peak at *m*/*z* 111.007 [M−H−CO_2_−2H_2_O]^−^, and was related to citric acid [18]. A key step in the dereplication of phenolic acid glycosides was the neutral losses of 162.05, 132.04, and 308.11 Da, corresponding to hexose, pentose, and rutinose, respectively, together with the base peaks of the respective phenolic acid deprotonated molecule. Thus, pentosylhexosides of hydroxybenzoic acid (**10**) and vanillic acid (**11**), dihexoside of caffeic acid (**14**), and hexoside of coumaric acid (**22**) were annotated. Compounds **13** ([M−H]^−^ at *m*/*z* 487.146) and **23** ([M−H]^−^ at *m*/*z* 501.161) gave base peaks at *m*/*z* 179.034 [caffeic acid−H]^−^ and 193.050 [ferulic acid−H]^−^, corresponding to the loss of rutinose ([M−H−rutinose]^−^), and were identified as caffeic acid *O*-rutinoside and ferulic acid *O*-rutinoside, respectively (Table 1, Figure 2). Compounds **10**, **11**, **12**, **14**, **15**, **16**, **17**, **18**, **19**, **22**, **23**, **24**, and **25** are reported for the first time in *S. scardica*.

#### 2.1.3. Caffeic Acids Oligomers

Caffeic acid oligomers consist of ester-bonded monomers such as danshensu, caffeic acid, and others and are present in Lamiaceae species [19,20,21]. Based on the accurate masses, MS/MS data, and literature data, a dimer rosmarinic acid (**29**), two trimers (**27** and **30**), and two tetramers (**28** and **31**) were dereplicated in the studied *S. scardica* extract. The fragmentation pattern and retention time of rosmarinic acid (**29**) were compared with reference standard. Key points in the caffeic acid oligomers annotation were the indicative fragment ions derived from the cleavage of *a* and *b* ester bonds with loss of danshensu [M−H−198.05]^−^, danshensoyl [M−H−180.04]^−^, and caffeoyl residue [M−H−162.03]^−^, respectively [19]. Compound **30** ([M−H]^−^ at *m*/*z* 491.099) afforded a base peak at *m*/*z* 311.056, corresponding to the easier loss of danshensu, due to the dibenzooxepin structure, restraining the cleavage of the *a* bond. Based on a comparison with literature data, **30** was tentatively identified as isosalvianolic acid C [19] (Table 1, Figure 3). With respect to **27**, an abundant fragment ion at *m*/*z* 493.114 [M−H−CO_2_]^−^ and a base peak at *m*/*z* 339.059 [M−H−198.05]^−^ were indicative of the presence of CO_2_ group attached to the benzofuran ring and danshensu residue. This allowed us to deduce the structure of lithospermic acid [19,20]. The fragmentation pathway of **28** included prominent ions at *m*/*z* 673.157 [M−H−CO_2_]^−^, 537.105 [M−H−180.04]^−^, 519.095 [M−H−198.05]^−^, 493.115 [M−H−180.04−CO_2_]^−^, 339.051 [M−H−2 × 198.05]^−^, and a base peak at *m*/*z* 321.041 [M−H−198.05−180.04−CO_2_]^−^, indicating, consequently, losses of two danshensu residues and carboxyl groups. Moreover, diagnostic ions at *m*/*z* 537.093, corresponding to deprotonated lythospermic acid, as well as the lack of loss of caffeoyl residue, suggested a terminal danshensu residue linked to lithospermic acid. Thus, compound **28** was dereplicated as salvianolic acid B (Table 1, Figure 3). Similarly, **31** was related to didehydrosalvianolic acid B (Table 1, Figure 3) [19]. Compounds 27, **28**, 30, and **31** are reported for the first time in *S. scardica*.

#### 2.1.4. Acylquinic Acid

Overall, three caffeoylquinic (**33**, **36**, and **38**), four *p*-coumaroylquinic (**35**, **41**, **42**, and **44**), two syringoylquinic (**37** and **40**), three feruloylquinic (**39**, **43**, and **45**), together with two hexosides (**32** and **34**), and a syringoyl–caffeoylquinic acid (**46**) were dereplicated or annotated in the studied extract (Table 1). The acylquinic acids (AQAs) annotation was based on the fragment ions and their relative abundances corresponding to each subclass AQAs [22,23,24]. Three isobars shared the same deprotonated molecule [M−H]^−^ at *m*/*z* 353.086. Compound **36** was identified as chlorogenic acid (5-caffeylquinic acid) due to the base peak at *m*/*z* 191.055 [quinic acid−H]^−^. The positional isomer neochlorogenic acid (3-caffeylquinic acid) (**33**) was apparent by the higher relative abundances of the fragment ions at *m*/*z* 179.033 (61.2%) and 135.043 (48.7%) than those of **36**. Compounds **33** and **36** were unambiguously identified by comparison with reference standards. In the MS/MS spectrum of **38**, **40**, and **42**, a base peak at *m*/*z* 173.044 [quinic acid−H−H_2_O]^−^ was detected, indicating caffeoyl, syringoyl, and *p*-coumaroyl residues at position 4 of the quinic acid. Thus, **38**, **40**, and **42** were annotated as 4-caffeylquinic, 4-syringoyl, and 4-*p*-coumaroyl acids, respectively [24]. The compounds 3-*p*-coumaroylquinic acid (3-*p*-CoQA) (**35**) and 3-feruloylquinic acid (3-FQA) (**39**) were identified from the base peaks at *m*/*z* 163.039 [*p*-CoA−H]^−^ and 193.050 [FA-H]^−^ (Table 1). Compounds **37**, **41**, and **43** showed a base peak at *m*/*z* 191.055 and fragment ions at *m*/*z* 197.045 [syringic acid−H]^−^, 163.038 [*p*-CoA−H]^−^, and 193.050 [FA−H]^−^, respectively, and were identified as 5-syringoylquinic, 5-*p*-coumaroylquinic, and 5-feruloylquinic (FQA) acids [22,23,24] (Table 1).

With respect to compound **46**, the base peak at 197.045 [syringic acid-H]^−^, together with a diagnostic fragment ion at *m*/*z* 335.077 [CQA−H−H_2_O]^−^ indicated syringoyl–caffeoylquinic acid. Additionally, two hexosides of neochlorogenic (**32**) and chlorogenic acid (**34**) were also dereplicated (Table 1). 

#### 2.1.5. Acylhexaric Acids

Key steps in the acylhexaric acids annotation were the subsequent losses of one hydroxydihydrocaffeoyl (**47**, **48**) and two (**49**, **50**) hydroxydihydrocaffeoyl and syringoyl residues (Table 1, Figure 4). Thus, the base peak in the MS/MS spectra was consistent with [hexaric acid (HA)−H]^−^ at *m*/*z* 209.030 (C_6_H_9_O_8_) supported by the series of indicative ions at *m*/*z* 191.019 [HA-H-H_2_O]^−^, 147.029 [HA−H−H_2_O−CO_2_]^−^, 129.018 [HA−H−2H_2_O−CO_2_]^−^, 111.007 [HA−H−3H_2_O−CO_2_]^−^, and 85.028 [HA−H−2H_2_O−2CO_2_]^−^ (Table 1) [25]. Compounds **49** and **50** shared the same [M−H]^−^ at *m*/*z* 569.116. They formed the prominent fragment ions at *m*/*z* 389.073 [M−H−180.04]^−^, 371.063 [M−H−198.05]^−^, and 209.030 [M−2 × 180.04]^−^, resulting from the concomitant loss of hydroxydihydrocaffeoyl and syringoyl residues. Syringoyl moiety was suggested by the fragment ions at *m*/*z* 197.045 [syringic acid (SA)−H]^−^, 182.021 [(SA−H)−CH_3_]•^−^ and 153.055 [(SA−H)−CO_2_]^−^. Compounds **49** and **50** were identified as isomeric hydroxydihydrocaffeoyl–syringoyl–hexaric acids (Table 1, Figure 4). Acylhexaric acids are reported for the first time in *S. scardica*.

#### 2.1.6. Phenylethanoid Glycosides

A class of secondary metabolites distinctive for *Sideritis* species were phenylethanoid glycosides [12]. The typical fragmentation pattern revealed the loss of 162.05, 146.05, 179.03, 18.01 Da—corresponding to glucosyl and rhamnosyl moieties—deprotonated caffeic acid, and H_2_O, respectively. Detailed discussion on the MS/MS fragmentation has been previously provided [12]. Based on a comparison with literature data, 10 phenylethanoid glycosides were dereplicated in the studied *S. scardica* extract (Table 1).

#### 2.1.7. Iridoid Glycosides

The characteristic loss of hexose (−162.05 Da) and 7-(hydroxymethyl)-4,5-dihydrocyclopentapyran-4,5-diol (−182.06 Da, C_9_H_10_O_4_) indicated the presence of iridoid glycosides [12]. Compound **61** with deprotonated molecules at *m*/*z* 523.166 was dereplicated as melittoside [12]. Fragment ions at *m*/*z* 163.039, 179.034, and 193.050 corresponding to the deprotonated coumaric, caffeic, and ferulic acids, led to the identification of **62**, **63**, and **64** as *p*-coumaroylmelittoside, caffeoylmelitoside, and feruloylmelitosside, respectively. Compounds **63** and **64** were found for the first time in *S. scardica* (Table 1). Previously, **63** has been isolated from *S. lanata* [26], while **64** has been found in *S. trojana* [27].

#### 2.1.8. Flavonoids

Flavonoids are the dominant secondary metabolites in *Sideritis* species [12]. The main flavonoids in the studied species were *O*-glycosides of the flavones isoscutellarein ([Agl−H]^−^ at *m*/*z* 285.041), methylisoscutellarein ([Agl−H]^−^ at *m*/*z* 299.056), hypolaetin ([Agl−H]^−^ at *m*/*z* 301.036), methylhypolaetin ([Agl−H]^−^ at *m*/*z* 315.052), and apigenin ([Agl−H]^−^ at *m*/*z* 285.041). The MS/MS fragmentation patterns of the glycosides have been described previously [12] and are based on the loss of 162.054, 324.106, and 42.016 Da, corresponding to O-hexose/dihexose/acetyl group, respectively. The occurrence of a 180 Da loss [M−H-hex-H_2_O]^−^ was indicative of 1→2 glycosylation between two sugars [16]. Significant fragments in the flavone aglycone annotation were a series of neutral losses of CO (−28 Da), CO_2_ (−44 Da), CH_2_O (−30 Da), and H_2_O (−18 Da), supported by the retro-Diels–Alder (rDA) cleavages ^0,4^A^−^, ^1,2^A^−^, ^1,3^A^−^, ^1,2^B^−^, and ^1,3^B^−^ [22,23]. In general, five glycosides of isoscutellarein (IS) (**70**, **71**, **72**, **78**, **83**), three of methylisoscutellarein (MIS) (**80**, **82**, **88**), two of hypolaetin (HL) (**67**, **73**), four of methylhypolaetin (MHL) (74, **76**, **79**, and **84**), five of apigenin (Api) (**65**, **69**, **75**, **77**, **86**), and one of luteolin (Lu) (**66**) were dereplicated in the *S. scardica* extract (Table 1). Apigenin and luteolin were evidenced by the rDA ions at *m*/*z* 151.002 (^1,3^A^−^), 107.012 (^0,4^A^−^), 117.033 (^1,3^B^−^) (Api), and 133.028 (^1,3^B^−^) (Lu). The aglycones isoscutellarein and hypolaetin were deduced from the rDA cleavages ^1,2^A^−^-H_2_O at *m*/*z* 163.003, ^1,3^A^−^-CO at *m*/*z* 136.986, as well as ^1,3^B^−^ at *m*/*z* 117.033 (IS) and ^1,3^B- at *m*/*z* 133.028 (HL). Their methoxylated derivatives revealed a fragment ion [Agl−H−CH_3_]•^−^ at *m*/*z* 284.033 (MIS) and 300.028 (MHL). Compound **70** ([M−H]^−^ at *m*/*z* 579.136) gave a base peak at *m*/*z* 285.041 [M−H−pent-hex]^−^. Thus, **70** was related to isoscutellarein *O*-pentosylhexoside. Illustrations of the fragmentation pathways of glycosides of the four abovementioned flavones aglycons are presented in Figure 5. The flavanone naringenin [M−H]^−^ at *m*/*z* 271.061 (**85**) was identified based on the RDA fragments at *m*/*z* 151.002 (^1,3^A^−^), 107.012 (^0,4^A^−^), and 119.049 (^1,3^B^−^). In addition, its dihexoside (**68**) and coumaroylhexoside (**87**) were annotated based on the neutral loss of two hexoses and coumaroylhexose, respectively. Compounds **68**, **70**, **85**, and **87** were annotated for the first time.

#### 2.1.9. Fatty Acids and Organosulfur Compounds

A saturated (**98**), two monounsaturated (**94** and **97**), and four polyunsaturated (**93**, **95**, **96**, and **99**) fatty acids were tentatively identified in *S. scardica* extract. Among them, the main fatty acids were trihydroxyoctadecadienoic acid (**93**) and trihydroxyoctadecenoic acid (**94**), previously reported for the species [12]. The dereplication and fragmentation pathway of dihydroxy fatty acids have been previously described [28]. In addition, four organosulfur compounds (**100**–**103**) were dereplicated based on fragment ions at *m*/*z* 96.959 [HO_4_S]^−^ and 79.956 [O_3_S]^−^. These compounds have been previously described for *S. scardica* [12].

**Table 1 molecules-29-00204-t001:** UHPLC–HRMS metabolite profiling of *Sideritis scardica* infusion with content (mg/g lyophilized infusion) of compounds assayed.

No	Identified/Tentatively Annotated Compound	Molecular Formula	Exact Mass [M−H]^−^	Fragmentation Pattern in (-) ESI-MS/MS	tR (min)	Δ ppm	Level of Confidence [29]	Content [mg/g li] Mean ± SD
**Sugar acids and saccharides**								
**1.**	Xylonic acid	C_5_H_10_O_6_	165.0405	165.0395 (60.5), 147.0288 (8.8), 129.0181 (12.3), 111.0073 (0.3), 105.0179 (33.1), 101.0228 (2.7), 87.0073 (43.1), 75.0072 (100)	0.68	−6.006	2	-
**2.**	Hexose	C_6_H_12_O_6_	179.0561	179.0541 (68.6), 161.0444 (11.1), 143.0338 (10.1), 125.0231 (1.53), 99.0437 (1.71), 81.0331 (4.8), 75.0072 (100)	0.70	−4.975	2	-
**3.**	Gluconic acid	C_6_H_12_O_7_	195.0510	195.0503 (96.1), 177.0396 (18.6), 159.0288 (14.7), 147.0287 (15.5), 141.0184 (5.9), 129.0180 (48.3), 111.0073 (5.3), 105.0179 (332.8), 75.0072 (100)	0.72	−3.619	2	-
**4.**	Pentose	C_5_H_10_O_5_	149.0456	149.0444 (21.6), 131.0334 (5.2), 101.0224 (0.8), 89.0229 (17.8), 75.0072 (100)	0.74	−7.358	2	-
**5.**	Asystoside	C_25_H_44_O_15_	583.2607	583.2618 (100), 421.2090 (4.9), 289.1662 (15.4), 161.0445 (15.8), 101.0230 (21.3), 71.0123 (30.6)	6.34	1.880	2	-
**Carboxylic acids**								
**6.**	Quinic acid	C_7_H_12_O_6_	191.0561	191.0553 (100), 173.0446 (1.8), 155.0340 (0.2), 137.022 (0.2), 127.0387 (3.3), 99.0438 (0.6), 93.0331 (5.6), 85.0279 (18.2), 71.0123 (1.6), 59.0123 (1.37)	0.69	−4.194	2	-
**7.**	Oxaloglutaric acid	C_7_H_8_O_7_	203.0197	203.0191 (100), 159.0292 (2.3), 141.0181 (27.8), 115.0022 (11.7), 97.0279 (97.1), 95.0123 (14.2), 79.0174 (11.2), 72.9915 (7.1), 71.0123 (50.6), 69.0330 (66.2)	0.88	−2.984	2	-
**8.**	Citric acid	C_6_H_8_O_7_	191.0197	191.0191 (2.6), 173.0084 (1.43), 154.9979 (0.7), 147.0286 (0.4), 129.0181 (6.1), 111.0074 (100), 101.0231 (0.7), 87.0073 (43.8), 85.0280 (27.0)	0.90	−3.119	2	-
**Hydroxybenzoic, hydroxycinnamic acids, and their derivatives**								
**9.**	Gallic acid	C_7_H_6_O_5_	169.0143	169.0133 (37.5), 125.0231 (100), 97.0280 (3.8), 69.0330 (4.9)	1.13	−5.660	1	3.45 ± 0.51
**10.**	Hydroxybenzoic acid *O*-pentosylhexoside ^a^	C_18_H_24_O_12_	431.1194	431.1200 (69.3), 299.0776 (2.3), 137.0232 (100), 93.0331 (73.0)	1.78	1.231	2	1.32 ± 0.13
**11.**	Vanillic acid *O*-pentosylhexoside ^a^	C_19_H_26_O_13_	461.1301	461.1310 (76.2), 329.0879 (1.3), 167.0340 (100), 152.0104 (52.4), 123.0438 (11.3), 108.0203 (27.4)	1.99	2.052	2	2.27 ± 0.35
**12.**	Protocatechuic acid ^a^	C_7_H_6_O_4_	153.0193	153.0182 (15.9), 123.0439 (0.1), 109.0281 (100), 81.0330 (1.4), 65.0380 (0.38)	2.01	−7.397	1	7.98 ± 0.54
**13.**	Caffeic acid *O*-rutinoside (swertiamacroside)	C_21_H_28_O_13_	487.1457	487.1465 (28.9), 179.0342 (100), 161.0234 (14.4), 135.0439 (45.5), 113.0232 (2.9)	2.80	1.572	2	4.68 ± 0.45
**14.**	Caffeic acid *O*-dihexoside ^a^	C_21_H_28_O_14_	503.1406	503.1414 (34.7), 341.0878 (42.9), 179.0341 (100), 135.0438 (77.1)	2.82	1.513	2	-
**15.**	2,3-Dihydroxybenzoic acid ^a^	C_7_H_6_O_4_	153.0193	153.0182 (51.9), 123.0074 (26.7), 108.0203 (100), 95.0124 (32.1), 85.0280 (33.1)	2.95	−7.267	2	-
**16.**	*p*-Hydroxybenzoic acid ^a^	C_7_H_6_O_3_	137.0244	137.0231 (35.2), 108.0200 (3.5), 93.0331 (100)	2.97	−9.614	1	-
**17.**	Dihydrocaffeic acid ^a^	C_9_H_10_O_4_	181.0506	181.0499 (52.1), 163.0377 (0.4), 137.0596 (100), 135.0439 (19.6), 123.0436 (58.0), 121.0282 (24.9), 119.0489 (15.1), 109.0281 (27.6), 93.0332 (2.5), 59.0124 (86.3)	3.33	−4.154	2	-
**18.**	2,4-Dihydroxybenzoic acid ^a^	C_7_H_6_O_4_	153.0193	153.0182 (76.8), 135.0075 (29.7), 123.0439 (0.26), 109.0281 (100), 108.0201 (0.2), 91.0174 (5.6), 81.0333 (0.3), 65.0381 (14.5)	3.47	−7397	1	4.77 ± 0.74
**19.**	*p*-Hydroxyphenyl acetic acid ^a^	C_8_H_8_O_3_	151.0400	151.0389 (100), 136.0156 (20.1), 123.0075 (4.6), 109.0283 (11.5), 107.0489 (2.3)	3.49	−7.133	1	-
**20.**	Caffeic acid	C_9_H_8_O_4_	179.0350	179.0341 (21.3), 135.0439 (100), 117.0335 (0.7), 107.0487 (1.35), 91.0537 (0.5)	3.54	−4.759	1	87.25 ± 6.54
**21.**	Gentisic acid	C_7_H_6_O_4_	153.0193	153.0182 (45.9), 109.0281 (100), 91.0175 (1.1), 108.0203 (8.7), 81.0330 (1.9), 65.0382 (0.1)	3.65	−7.397	1	0.42 ± 0.05
**22.**	Ferulic acid *O*-rutinoside ^a,b^	C_22_H_30_O_13_	501.1614	501.1618 (10.2), 193.0500 (100), 175.0393 (8.4), 160.0157 (9.7), 134.0361 (43.4), 113.0230 (5.5)	3.78	3.125	2	-
**23.**	*p*-Coumaric acid *O*-hexoside ^a^	C_15_H_18_O_8_	325.0928	163.0391 (100), 145.0288 (4.14), 119.0488 (35.7)	3.81	3.197	2	-
**24.**	Syringic acid ^a^	C_9_H_10_O_5_	197.0455	197.0449 (40.0), 182.0213 (48.9), 153.0547 (100), 138.0311 (11.5), 123.0075 (26.7), 121.0281 (85.6), 106.0046 (9.67), 95.0123 (8.1), 89.0018 (14.97)	4.34	−3.231	2	1.69 ± 0.26
**25.**	*O*-coumaric acid ^a^	C_9_H_8_O_3_	163.0401	163.0390 (9.7), 135.0075 (0.2), 119.0489 (100)	4.55	−6.363	1	-
**26.**	Ferulic acid	C_10_H_10_O_4_	193.0506	193.0499 (100), 178.0260 (14.1), 165.0545 (13.5), 149.0600 (21.4), 134.0358 (12.3), 123.0438 (92.0), 79.0538 (4.1)	5.17	−3.637	1	-
**Caffeic acid oligomers**								
**27.**	Lithospermic acid ^a^	C_27_H_22_O_12_	537.1038	537.1035 (7.1), 493.1142 (30.4), 339.0515 (100), 313.0726 (8.8), 295.0613 (23.6), 267.0671 (10.9), 179.0345 (12.4), 135.0440 (49.3)	4.97	−0.743	2	0.51 ± 0.05
**28.**	Salvianolic acid B ^a^	C_36_H_30_O_16_	717.1460	717.1478 (49.4), 673.1570 (7.7), 537.1055 (28.0), 519.0946 (47.9), 493.1153 (5.9), 339.0512 (13.0), 321.0409 (26.0), 313.0717 (11.5), 295.0616 (100), 277.0513 (6.7), 229.0141 (8.7), 203.0346 (13.9), 197.0447 (2.5), 179.0340 (11.3), 161.0237 (1.1), 135.0439 (32.2), 109.0281 (71.9)	6.06	2.401	2	1.42 ± 0.01
**29.**	Rosmarinic acid	C_18_H_16_O_8_	359.0772	359.0782 (14.1), 197.0449 (26.0), 179.0342 (11.3), 161.0233 (100), 135.0439 (14.64), 133.0282 (19.8), 109.0275 (0.4)	6.33	2.56	1	6.07 ± 0.46
**30.**	Isosalvianolic acid C ^a^	C_26_H_20_O_10_	491.0983	491.0991 (100), 311.0567 (98.4), 267.0666 (39.3), 265.0508 (3.9), 249.0559 (1.5), 197.0454 (2.2), 179.0339 (1.8), 135.0440 (48.7)	7.29	1.466	2	0.42 ± 0.03
**31.**	Didehydrosalvianolic acid B ^a^	C_36_H_28_O_16_	715.1305	715.1324 (52.4), 535.0894 (20.2), 517.0786 (5.4), 337.0357 (8.1), 319.0241 (12.4), 311.0575 (7.6), 293.0461 (100), 265.0503 (7.1), 197.0446 (8.9), 135.0438 (10.5), 109.0279 (5.5)	7.76	2.786	2	0.19 ± 0.03
**Acylquinic acids**								
**32.**	(Neo)chlorogenic acid *O*-hexoside ^a^	C_22_H_28_O_14_	515.1406	515.1414 (52.2), 353.0883 (5.6), 191.0553 (100), 179.0351 (3.4), 135.0441 (4.0), 93.0332 (11.0)	2.13	1.536	2	-
**33.**	Neochlorogenic acid	C_16_H_18_O_9_	353.0877	353.0883 (40.7), 191.0555 (100), 179.0341 (61.2), 173.0451 (3.1), 161.0235 (3.5), 135.0439 (48.7), 93.0331 (4.5)	2.36	0.575	1	-
**34.**	Chlorogenic acid *O*-hexoside ^a^	C_22_H_28_O_14_	515.1406	515.1414 (100), 323.0767(51.9), 191.0554 (94.8), 179.0327 (4.5), 161.0238 (33.5), 135.0434 (6.4), 111.0435 (4.0)	2.85	2.487	1	-
**35.**	3-*p*-coumaroylquinic acid ^a^	C_16_H_18_O_8_	337.0928	337.0932 (6.9), 191.0555 (6.9), 173.0449 (3.5), 163.0390 (100), 135.0437 (0.5), 119.0488 (26.5), 111.0438 (0.7), 93.0332 (0.8)	2.99	0.918	2	-
**36.**	Chlorogenic acid	C_16_H_18_O_9_	353.0877	353.0879 (4.6), 191.0554 (100), 179.0380 (0.9), 161.0235 (1.9), 111.0437 (0.8), 93.0331 (2.7), 85.0280 (7.5)	3.17	0.325	1	5.22 ± 0.21
**37.**	5-Syringoylquinic acid ^a^	C_16_H_20_O_10_	371.0983	371.0987 (36.1), 197.0448 (4.9), 191.0554 (100), 173.0443 (14.6), 153.0538 (2.3), 121.0279 (9.4), 111.0435 (2.8), 93.0331 (31.6), 85.0279 (3.7)	3.30	0.862	3	-
**38.**	4-Caffeoylquinic acid ^a^	C_16_H_18_O_9_	353.0877	353.0883 (32.7), 191.0555 (39.50), 179.0342 (75.3), 173.0447 (100), 135.0439 (52.1), 111.0436 (3.3), 93.0331 (21.3)	3.36	0.515	2	7.65 ± 0.96
**39.**	3-Feruloylquinic acid	C_17_H_20_O_9_	367.1034	367.1036 (19.5), 193.0500 (100), 173.0447 (4.7), 137.0226 (3.4), 134.0361 (56.7)	3.42	0.339	2	0.63 ± 0.04
**40.**	4-Syringoylquinic acid ^a^	C_16_H_20_O_10_	371.0983	371.0978 (9.8), 197.0456 (11.5), 191.0554 (100), 173.0443 (14.6), 153.0538 (2.3), 121.0283 (4.9), 111.0435 (2.8), 93.0331 (14.5)	3.43	−1.509	3	-
**41.**	5-*p*-Coumaroylquinic acid ^a^	C_16_H_18_O_8_	337.0928	337.0940 (8.9), 191.0554 (100), 173.0446 (6.6), 163.0390 (5.31), 119.0488 (4.8), 111.0437 (2.0), 93.0331 (16.7)	3.96	3.172	2	1.85 ± 0.03
**42.**	4-*p*-Coumaroylquinic acid ^a^	C_16_H_18_O_8_	337.0928	337.0941 (8.6), 191.0555 (2.7), 173.0445 (100), 163.0390 (18.5), 119.0488 (9.0), 111.0436 (3.1), 93.0330 (22.2)	4.02	3.558	2	1.82 ± 0.05
**43.**	5-Feruloylquinic acid	C_17_H_20_O_9_	367.1034	367.1038 (23.7), 193.0500 (16.9), 191.0554 (100), 173.0446 (81.4), 155.0340 (3.8), 134.0361 (19.2), 111.0438 (4.9), 93.0331 (39.7)	4.38	0.912	2	5.13 ± 0.08
**44.**	1-*p*-Coumaroylquinic acid ^a^	C_16_H_18_O_8_	337.0928	337.0932 (7.2), 191.0554 (100), 173.0445 (100), 135.0447 (2.3), 163.0393 (0.5), 119.0487 (1.1), 111.0437 (0.5), 93.0331 (5.3)	4.60	1.007	2	0.56 ± 0.04
**45.**	1-Feruloylquinic acid	C_17_H_20_O_9_	367.1034	367.1039 (10.2), 191.0554 (100), 179.0340 (0.5), 173.0448 (2.3), 161.0239 (0.3), 134.0360 (3.1), 111.0440 (1.41), 93.0331 (5.15)	4.90	−2.996	2	1.12 ± 0.16
**46.**	Syringoyl–caffeoylquinic acid ^a^	C_25_H_26_O_13_	533.1300	533.1305 (14.5), 335.0777 (44.6), 291.0875 (20.7), 197.0450 (100), 153.0546 (12.6), 137.0232 (38.4), 123.0073 (17.0), 111.0439 (10.4), 93.0331 (49.0)	6.90	0.799	2	0.81 ± 0.08
**Acylhexaric acids**								
**47.**	Hydroxydihydrocaffeoyl–hexaric acid ^a,b^	C_15_H_18_O_12_	389.0725	389.0730 (16.8), 371.0631 (0.8), 209.0297 (16.8), 191.0192 (34.2), 153.0539 (1.0), 147.0286 (17.9), 129.0180 (10.8), 85.0280 (100)	1.65	1.236	3	1.12 ± 0.16
**48.**	Hydroxydihydrocaffeoyl–hexaric acid isomer ^a,b^	C_15_H_18_O_12_	389.0725	389.0728 (16.8), 371.0621 (2.7), 209.0294 (17.2), 197.0442 (3.7), 191.0191 (45.3), 173.0087 (3.1), 147.0286 (20.6), 129.0180 (13.5), 111.0079 (4.5), 85.0280 (100)	2.60	0.773	3	0.81 ± 0.08
**49.**	Hydroxydihydrocaffeoyl–siryngoyl–hexaric acid ^a,b^	C_24_H_26_O_16_	569.1148	569.1159 (49.1), 389.0726 (8.8), 371.0627 (52.7), 327.0726 (13.8), 209.0299 (1.4), 197.0450 (39.1), 191.0186 (3.7), 182.0211 (3.3), 173.0084 (18.4), 166.9975 (1.6), 147.0285 (10.0), 138.0309 (1.7), 129.0181 (54.5), 123.0072 (1.6), 121.0282 (17.5), 111.0073 (14.3), 97.6908 (1.7), 85.0280 (100)	3.53	1.937	3	1.77 ± 0.15
**50.**	Hydroxydihydrocaffeoyl–siryngoyl–hexaric acid isomer ^a,b^	C_24_H_26_O_16_	569.1148	569.1158 (43.9), 389.0731 (7.6), 371.0625 (42.8), 327.0728 (12.2), 209.0295 (1.13), 197.0450 (40.8), 191.0186 (3.8), 182.0214 (4.3), 173.0084 (15.4), 166.9978 (1.0), 153.0548 (10.2), 147.0288 (8.0), 138.0315 (2.2), 129.0181 (59.1), 123.0074 (3.6), 121.0281 (16.5), 111.0073 (12.5), 85.0280 (100)	3.77	2.148	3	3.53 ± 0.237
**Phenylethanoid glycosides**								
**51.**	Decaffeoyl aceteoside/verbasoside	C_20_H_30_O_12_	461.1664	461.1673 (100), 315.1085 (5.2), 297.0984 (2.6), 135.0439 (29.4), 113.0230 (46.0), 85.0280 (20.6), 71. 0123 (20.7)	2.57	0.881	2	35.09 ± 2.46
**52.**	Hydroxyverbascoside	C_29_H_36_O_16_	639.1931	639.1946 (86.8), 621.1832 (5.3), 459.1533 (1.9), 179.0342 (36.0), 161.0232 (5.3), 135.0440 (22.3), 133.0283 (37.6), 113.0231 (8.6)	4.49	2.381	2	13.45 ± 0.47
**53.**	Verbascoside	C_29_H_36_O_15_	623.1987	623.1996 (60.3), 461.1674 (9.3), 315.1078 (1.80), 179.0340 (2.4), 161.0234 (100), 135.0440 (8.7), 133.0283 (23.9)	5.48	2.305	2	151.54 ± 10.86
**54.**	Echinacoside	C_35_H_46_O_20_	785.2509	785.2529 (72.5), 623.2175 (7.7), 461.1666 (6.3), 179.0349 (3.6), 161.0234 (100), 135.0440 (24.9), 133.0282 (47.0)	5.23	2.424	2	0.74 ± 0.005
**55.**	Forsythoside B/samioside/lavandulifolioside	C_34_H_44_O_19_	755.2403	755.2424 (80.5), 593.2080 (8.2), 461.1675 (8.7), 267.1614 (1.5), 179.0341 (6.8), 161.0234 (100), 135.0439 (23.5), 133.0282 (45.8), 113.0231 (9.4)	5.38	2.606	2	6.57 ± 0.103
**56.**	Alyssonoside	C_35_H_46_O_19_	769.2560	769.2578 (100), 593.2076 (8.5), 461.1658 (8.9), 193.05 (20.9), 175.0393 (44.1), 161.0236 (10.2), 160.0156 (44.4), 135.0442 (15.4), 134.0362 (20.9), 132.0206 (14.9), 123.0439 (7.4), 113.023 (15.1), 85.0281 (9.3), 71.0124 (10.0)	6.07	22.207	2	2.16 ± 0.14
**57.**	Leucoseptoside A	C_30_H_38_O_15_	637.2138	637.2154 (100), 461.1669 (16.2), 315.1091 (3.8), 193.0501 (13.00), 175.0392 (67.8), 161.0233 (109), 160.0155 (60.1), 113.0230 (18.9)	6.25	2.505	2	22.80 ± 0.82
**58.**	Leontoside B/stachyoside D	C_36_H_48_O_19_	783.2716	783.2734 (100), 193.0501 (34.7), 175.0392 (92.4), 167.0700 (1.5), 160.0156 (73.8), 132.0205 (26.8)	6.97	2.142	2	0.90 ± 0.05
**59.**	Acetylverbascoside	C_31_H_38_O_16_	665.2087	665.2103 (65.0), 503.1778 (3.3), 461.1676 (3.1), 161.0234 (100), 179.0339 (2.9), 135.0440 (10.4), 133.0283 (38.4), 113.0229 (1.9)	6.99	2.348	2	0.90 ± 0.01
**60.**	Martynoside	C_31_H_40_O_15_	651.2294	651.2317 (95.8), 475.1836 (1.2), 329.1252 (1.1), 193.0500 (13.6), 175.0392 (100), 160.0156 (69.1), 132.0204 (25.1), 113.0230 (14.0)	7.21	3.450	2	11.35 ± 0.75
**Iridoid glycosides**								
**61.**	Melittoside	C_21_H_32_O_15_	523.1668	523.1676 (10.9), 361.1139 (4.5), 343.1040 (5.4), 325.0919 (1.3), 313.9523 (0.4), 283.0820 (0.5), 253.0722 (1.3), 223.0613 (1.7), 205.0506 (1.6), 179.0553 (100), 161.0447 (16.8), 119.0337 (34.9), 101.0230 (30.8), 89.0229 (80.9)	1.33	1.446	2	13.22 ± 1.36
**62.**	*p*-Coumaroylmelittoside	C_30_H_38_O_17_	669.2035	669.2049 (100), 489.1423 (5.4), 325.0932 (28.3), 307.0823 (5.0), 265.0729 (3.2), 235.0605 (3.4), 205.0499 (16.9), 163.0390 (71.9), 145.0283 (84.4), 119.0488 (46.4), 93.0330 (4.9), 89.0230 (6.4)	3.98	1.909	2	3.71 ± 0.06
**63.**	Caffeoylmelittoside ^a,b^	C_30_H_38_O_18_	685.1974	685.1998 (100), 649.1036 (1.3), 523.1473 (1.9), 187.0396 (1.1), 181.0498 (24.9), 179.0342 (61.7), 163.0391 (39.9), 161.0235 (12.7), 135.0440 (78.4), 93.0331 (2.4), 89.0227 (2.3)	4.79	1.799	2	0.70 ± 0.01
**64.**	Feruloylmelittoside ^a,b^	C_31_H_40_O_18_	699.2142	699.2161 (75.9), 519.1519 (8.4), 357.0992 (54.0), 193.0500 (100), 163.0389 (30.6), 135.0435 (16.10), 134.0361 (70.0)	5.56	2.778	2	-
**Flavonoids**								
**65.**	Apigenin 6,8-*C*-hexosyl hexoside	C_27_H_30_O_15_	593.1522	593.1522 (100), 503.1212 (4.8), 473.1094 (14.5), 413.0891 (2.0), 395.0770 (0.8), 383.0776 (18.0), 353.0672 (30.1), 325.0729 (2.8), 297.0770 (10.7), 161.0235 (2.2), 117.0333 (3.3)	4.05	1.630	2	1.59 ± 0.18
**66.**	Luteolin 7-*O*-dihexoside	C_27_H_30_O_16_	609.1461	609.1473 (100), 447.0949 (4.7), 285.0408 (61.9), 284.0331 (18.1), 256.0380 (1.0), 151.0025 (3.8), 133.0281 (3.4), 107.0121 (2.1)	4.94	2.023	2	0.58 ± 0.01
**67.**	Hypolaetin 7-*O*-hexosyl (1→2)-hexoside	C_27_H_30_O_17_	625.1410	625.1422 (100), 463.0883 (6.6), 445.0775 (6.5), 301.0356 (88.8), 300.0279 (30.4), 283.0244 (0.7), 255.0292 (3.6), 227.0350 (2.3), 166.9975 (3.3), 163.0029 (1.1), 137.0232 (4.0), 133.0280 (7.5)	5.08	1.900	1	5.30 ± 0.09
**68.**	Naringenin 7-*O*-dihexoside	C_27_H_32_O_15_	595.1668	595.1658 (100), 475.1172 (4.6), 355.0674 (5.6), 271.0619 (54.8), 270.0215 (1.8), 269.0457 (38.2), 151.0027 (43.4), 119.0488 (21.5), 107.0123 (18.3)	5.45	−1.703	2	-
**69.**	Apigenin 7-*O*-allosyl (1→2) glucoside	C_27_H_30_O_15_	593.1512	593.1522 (64.7), 431.0986 (3.0), 269.0459 (100), 225.0546 (0.7), 151.0027 (1.4), 161.0232 (1.7), 117.0333 (3.3), 107.0125 (1.8)	5.50	1.731	2	1.93 ± 0.04
**70.**	Isoscutellarein 7-*O*-pentosyl–hexoside ^a,b^	C_26_H_28_O_15_	579.1356	579.1366 (38.9), 461.0086 (0.4), 285.0408 (100), 257.0451 (1.7), 241.0506 (1.6), 229.0497 (1.5), 213.0554 (5.1), 187.0393 (4.4), 136.9867 (0.9), 117.0330 (0.5)	5.51	1.825	2	1.01 ± 0.05
**71.**	Isoscutellarein 7-*O*-hexosyl (1→2)-hexoside	C_27_H_30_O_16_	609.1461	609.1472 (87.2), 447.0912 (0.4), 429.0833(6.8), 285.0409 (100), 284.0328 (9.5), 255.0303 (2.2), 167.0496 (0.5), 163.0028 (2.1), 136.9868 (1.7), 117.0333 (1.8)	5.62	1.826	2	47.47 ± 0.95
**72.**	Isoscutellarein 7-*O*-hexoside	C_21_H_20_O_11_	447.0933	447.0941 (26.4), 285.0408 (100), 229.0509 (1.0), 136.9868 (2.3), 117.0332 (1.1)	5.81	1.801	2	0.58 ± 0.01
**73.**	Hypolaetin 7-*O*-acetylhexosiyl–hexoside	C_29_H_32_O_18_	667.1516	625.1406 (3.2), 463.0898 (3.2), 445.079 (6.6), 301.0357 (76.0), 300.0278 (30.8), 283.0285 (1.1), 255.0298 (1.4), 227.0347 (1.0), 166.9973 (1.9), 163.0020 (1.0), 137.0232 (4.0), 133.0284 (8.3), 109.0282 (1.0)	5.93	3.152	2	19.17 ± 1.07
**74.**	Methylhypolaetin 7-*O*-dihexoside	C_28_H_32_O_17_	639.1567	639.1579 (75.3), 315.0516 (100), 300.0279 (40.7), 271.0264 (1.2), 243.0296 (2.5), 165.9901 (0.7), 136.9871 (6.2), 117.1944 (0.5), 133.0283 (2.5)	5.96	1.920	2	17.33 ± 0.25
**75.**	Apigenin 7-*O*-glucoside	C_21_H_20_O_10_	431.0984	431.0988 (100), 269.0453 (24.9), 268.0381 (54.3), 211.0395 (1.6), 151.0025 (3.2), 117.0330 (1.8), 170.0124 (2.0)	6.06	1.044	1	0.46 ± 0.02
**76.**	Methylhypolaetin 7-*O*-hexoside	C_22_H_22_O_12_	477.1039	477.1045 (30.5), 315.0515 (100), 300.0278 (32.0), 227.0350 (1.6), 136.9870 (6.2)	6.13	1.406	2	0.26 ± 0.01
**77.**	Apigenin 7-*O*-[6‴-*O*-acetyl]-hexosyl(1→2)-hexoside	C_29_H_32_O_16_	635.1618	635.1629 (60.6), 593.1563 (1.1), 431.0981 (2.3), 269.0458 (100), 225.0560 (12.4), 151.0024 (1.7), 117.0332 (5.1), 107.0126 (2.6)	6.44	1.798	2	2.18 ± 0.003
**78.**	Isoscutellarein 7-*O*-hexosyl-(1→2)-[6″-*O*-acetyl]-hexoside	C_29_H_32_O_17_	651.1567	651.1581 (69.3), 429.0831 (8.9), 285.0408 (100), 255.0285 (1.0), 239.0344 (1.1), 163.0026 (1.5), 136.9863 (0.8), 117.0334 (1.1)	6.58	2.161	2	151.70 ± 14.79
**79.**	4′-Methylhypolaetin 7-O-acetyl–hexosyl–hexoside	C_30_H_34_O_18_	681.1672	681.1688 (89.6), 639.1533 (1.2), 357.0594 (0.9), 315.0516 (100), 300.0279 (44.2), 271.0248 (1.6), 243.0291 (1.4), 136.9868 (7.9), 133.0283 (4.1)	6.83	2.236	2	78.33 ± 3.29
**80.**	4′-Methylisoscutellarein 7-*O*-dihexoside	C_28_H_32_O_16_	623.1618	623.1630 (100), 461.1117 (0.6), 299.0565 (83.6), 284.0330 (39.2), 255.0299 (3.0), 117.0330 (0.7)	7.23	1.929	2	24.20 ± 0.98
**81.**	Tremasperin	C_30_H_34_O_16_	649.1774	649.1786 (5.4), 607.1672 (2.6), 283.0616 (100), 268.0381 (55.5), 284.0649 (5.8), 240.0431 (2.2), 151.0024 (0.5)	8.21	1.913	2	0.55 ± 0.04
**82.**	4′-*O*-methylisoscutellarein 7-*O*-[6‴-*O*-acetyl]hexosyl-(1→2)hexoside	C_30_H_34_O_17_	665.1723	665.1740 (84.8), 299.0565 (100), 284.0330 (30.6), 255.0293 (2.5), 240.0429 (2.5), 227.0343 (2.5), 163.0025 (1.1), 136.9867 (9.2), 117.0338 (1.9)	8.24	2.447	2	107.44 ± 9.07
**83.**	Isoscutellarein 7-*O*-acetylhexosyl-*O*-acetylhexoside	C_31_H_34_O_18_	693.1672	693.1663 (85.8), 471.0903 (7.4), 285.0407 (100), 213.0551 (6.0), 163.0022 (4.5), 136.9864 (1.9), 117.0331 (3.4)	8.26	−1.323	2	-
**84.**	Methylhypolaetin 7-*O*-acetylhexosyl-*O*-acetylhexoside	C_32_H_36_O_19_	723.1778	723.1794 (89.7), 315.0515 (100), 300.0280 (44.9), 271.0255 (1.1), 243.0298 (1.4), 199.0390 (4.6), 136.9866 (9.5), 133.0284 (5.9)	8.47	2.182	2	0.31 ± 0.02
**85.**	Naringenin	C_15_H_12_O_5_	271.0612	271.0615 (100), 227.0701 (0.7), 165.0180 (2.5), 151.0025 (65.7), 125.0228 (1.3), 119.0489 (52.3), 107.0124 (15.5), 93.0331 (11.9)	8.60	1.193	2	-
**86.**	Apigenin 7-*O*-p-coumaroyl-*O*-hexoside	C_30_H_26_O_12_	577.1352	577.1360 (100), 431.0990 (13.4), 413.0890 (7.9), 269.0459 (77.0), 145.0283 (83.4), 163.0391 (4.4), 117.0332 (38.3), 107.0121 (1.1), 151.0026 (2.1)	9.06	1.457	2	0.75 ± 0.01
**87.**	Naringenin 7-*O*-coumaroylhexoside ^a^	C_30_H_28_O_12_	579.1508	579.1514 (100), 415.1033 (3.9), 307.0829 (12.0), 271.0616 (79.4), 151.0026 (40.9), 163.0391 (15.7), 145.0283 (57.4), 119.0489 (40.9), 107.0125 (15.2), 117.0332 (21.9)	9.15	0.985	2	1.09 ± 0.09
**88.**	4′-Methylisoscutellarein 7-*O*-(6‴-acetyl)-hexosyl(1→2)-[6′-*O*-acetyl]hexoside	C_32_H_36_O_18_	707.1829	707.1844 (12.1), 299.0565 (100), 284.0330 (31.9), 300.0598 (6.8), 298.0496 (8.2), 255.0292 (4.7), 240.0424 (3.2), 227.0341 (1.7), 163.0023 (3.5), 136.9867 (10.7), 117.0332 (1.5)	9.89	2.125	2	0.09 ± 0.003
**89.**	Pectolinarigenin	C_17_H_14_O_6_	313.0718	313.0721 (100), 298.0485 (53.9), 283.0251 (52.0), 269.0468 (2.4), 255.0302 (14.4), 227.0342 (2.17), 211.0386 (1.3), 183.0446 (2.1), 178.9918 (3.0), 163.0031 (11.8), 135.0075 (2.8), 117.0331 (13.1)	10.36	0.922	1	3.11 ± 0.50
**90.**	Eupatilin	C_18_H_16_O_7_	343.0823	343.0826 (85.2), 328.0594 (100), 313.0360 (54.6), 298.0125 (13.3), 285.0412 (2.0), 270.0174 (42.6), 257.0095 (2.6), 133.0282 (3.7), 123.0439 (4.6)	11.05	0.915	2	0.47 ± 0.07
**91.**	8-Methoxycirsilineol	C_18_H_16_O_7_	343.0823	343.0826 (100), 328.0594 (48.7), 313.0360 (71.8), 299.0952 (0.7), 298.0124 (209), 270.0175 (10.6), 242.0220 (4.2), 161.0233 (0.8), 117.0333 (8.9)	11.24	0.828	2	14.15 ± 2.30
**92.**	Genkwanin	C_16_H_12_O_5_	283.0611	283.0615 (100), 268.0379 (67.4), 240.0428 (6.2), 239.0352 (1.8), 178.9915 (1.2), 151.0025 (4.1), 107.0125 (3.3)	11.42	1.036	2	-
**Fatty acids**								
**93.**	Trihydroxyoctadecadienoic acid	C_18_H_32_O_5_	327.2177	327.2166 (100), 309.2069 (0.8), 291.1971 (3.5), 229.1443 (12.5), 211.1334 (16.2), 183.1383 (1.6), 171.1015 (6.0), 85.0280 (2.5), 57.0329 (0.9)	9.15	0.986	2	-
**94.**	Trihydroxyoctadecenoic acid	C_18_H_34_O_5_	329.2334	329.2338 (100), 311.2232 (1.4), 293.2119 (0.4), 229.1442 (17.2), 211.1335 (23.2), 183.1381 (2.7), 171.1020 (4.4), 127.1115 (1.6)	9.80	1.466	2	-
**95.**	Dihydroxyoctadecatrienoic acid	C_18_H_30_O_4_	309.2071	309.2076 (100), 291.1972 (54.3), 247.2075 (1.0), 185.1179 (5.4), 137.0959 (17.9), 97.0645 (4.1)	10.90	1.641	2	-
**96.**	Dihydroxyoctadecadienoic acid	C_18_H_32_O_4_	311.2228	311.2233 (100), 293.2132 (7.2), 275.2029 (6.4), 201.1128 (61.0), 183.1387 (1.6), 171.1015 (11.2), 127.1114 (4.3)12.62	12.62	1.662	2	-
**97.**	Dihydroxyoctadecenoic acid	C_18_H_34_O_4_	313.2384	313.2389 (100), 295.2266 (5.8), 277.2166 (4.9), 201.1126 (42.8), 171.1012 (5.11), 127.1116 (4.8), 125.0960 (3.3)	13.75	1.460	2	-
**98.**	Dihydroxyoctadecanoic acid	C_18_H_36_O_4_	315.2541	315.2544 (100), 297.2450 (4.3), 287.2241 (4.3), 171.1380 (0.6), 141.1272 (3.2), 127.1116 (0.6), 89.0230 (0.5)	14.87	1.101	2	-
**99.**	Hydroxylinoleic acid	C_18_H_32_O_3_	295.2279	295.2282 (100), 277.2175 (14.6), 195.1384 (17.5), 113.0960 (1.1)	15.98	1.056	2	-
**Organosulfur compounds**								
**100.**	Dodecyl sulfate	C_12_H_26_O_4_S	265.1479	265.1482 (100), 96.9586 (66.4), 79.9558 (1.6)	14.55	1.081	2	-
**101.**	Lauryl ether sulfate	C_14_H_30_O_5_S	309.1741	309.1746 (100), 122.9746 (1.9), 104.9527 (0.2), 96.9586 (54.3), 79.9558 (6.6)	16.10	1.624	2	-
**102.**	4-Dodecylbenenesulfonic acid	C_18_H_30_O_3_S	325.1842	325.1846 (100), 216.0095 (0.2), 183.0113 (46.5), 197.0272 (0.8), 184.0147 (1.9)	17.44	0.957	2	-
**103.**	Myristyl sulfate	C_14_H_30_O_4_S	293.1792	293.1796 (100), 96.9586 (73.6), 79.9558 (2.2)	17.89	1.455	2	-

^a^—reported for the first time in the studied species; ^b^—undescribed in the literature; level of confidence: 1—compound identified by comparison with reference standard; 2—putatively annotated compound; 3—putatively characterized compound classes.

### 2.2. Quantitative Determination

The quantitative determination of the main compounds in the profile of *S. scardica* lyophilized infusion was based on a common approach, where the HPLC analysis of the analytes was performed with a mobile phase composed of formic acid acetonitrile and water [22]. The content of the assayed compounds is revealed in Table 1. The main compounds in the tested lyophilized infusion were isoscutellarein-7-*O*-hexosyl-(1→2)-[6″-O-acetyl]-hexoside (**78**) and verbascoside (**53**), followed by 4′-*O*-methylisoscutellarein-7-*O*-[6‴-*O*-acetyl]hexosyl-(1→2)hexoside (**82**). Other dominant phenolic compounds include caffeic acid (**20**), 4′-methylhypolaetin-7-*O*-acetyl–hexosyl–hexoside (**79**), and isoscutellarein 7-*O*-hexosyl (1→2)-hexoside (**71**) (Table 1). Moreover, the data reveled moderate quantity of the phenylethanoid glycosides leucoseptoside A (**57**) and martynoside (**60**), as well as iridoid glycoside melittoside (**61**) (Table 1). The content of phenylethanoid glycosides ranged from 0.74 mg/g (**54**) to 151.54 mg/g lyophilized infusion (li) (**53**). With respect to caffeic acid oligomers, their quantities were found to range from 0.19 ± 0.033 mg/g li (**31**) to 6.07 ± 0.46 mg/g li (**29**), while caffeoylhexaric acids ranged from 0.81 ± 0.075 (**48**) to 3.53 ± 0.237 (**50**) (Table 1). However, this is the first attempt to quantify the secondary metabolites of the above classes in *Sideritis* species. 

4-Caffeoylquinic acid (**38**) was found to be the dominant acylquinic acid (7.65 ± 0.96 mg/g li), followed by chlorogenic (**36**) and 5-feruloylquinic acid (**43**). Earlier quantitative research on the genus *Sideritis* showed that 5-caffeoylquinic acid has been found in all studied species as the most abundant hydroxycinnamic acid. In addition, the dominant phenolic compounds were the isoscutellarein derivatives isoscutellarein 7-*O*-[6‴-*O*-acetyl]-allosyl-(1→2)glucoside and 4′-*O*-methylisoscutellarein 7-*O*-allosyl-(1→2)-[6″-O-acetyl]-glucoside. Recently, eight compounds were detected in different *Sideritis* species: 5-caffeoylquinic acid, lavandulifolioside, verbascoside, isoscutellarein 7-O-allosyl(1→2)glucoside, hypolaetin 7-O-[6″-O-acetyl]-allosyl(1→2)glucoside, isoscutellarein 7-O-[6″-O-acetyl]-allosyl(1→2) glucoside, 3′-O-methylhypolaetin 7-O-[6″-O-acetyl]-allosyl(1→2)glucoside, 4′-O-methylhypolaetin, and 7-O-[6″-O-acetyl]-allosyl-(1→2)-[6″-O-acetyl]-glucoside). They represent 50% to 80% of the total phenolic content in *S. scardica*, *S. raeseri*, *S. syriaca*, and *S. Taurica*, and up to 90% in *S. lanata* [6]. The most abundant compounds present in the analyzed *Siderits* samples belonged to the group of phenylethanoid glycosides. The content of phenylethanoid glycosides ranged from 1.22 mg/g dry herb for *S. lanata* to 108.3 mg/g dry herb for *S. scardica* from Rhodopi Mountain, Bulgaria. The contribution of phenylethanoid glycosides to total phenolic content was around 50% for all samples, except for *S. lanata* where it accounted only for around 7% [6]. Eleven acetylated glycosides of isoscutellarein, hypolaetin, methylhypolaetin and methylisoscutellarein were previously isolated from 80% EtOH extract [30]. The differences between previous studies and our results can be ascribed to the different extraction methods and solvents.

### 2.3. Study Strength, Limitation and Future Direction

The study strength is that the presented extraction method of infusion is similar to the approach used in traditional medicine to process *S. scardica* tea. Therefore, this provides an insight into the phytochemical composition of common tea used in the traditional medicine and in-home remedies. A notable contribution of this study is the first-time dereplication and fragmentation patterns of five caffeic acids oligomers and four acylhexaric acids in *S. scardica*, expanding the current understanding of its chemical profile. The quantitative analysis identified major compounds in *S. scardica* infusion, with phenylethanoid verbascoside, glycosides of isoscutellarein, methylisoscutelarein, hypolaetin, and caffeic acid standing out as significant constituents. The reported concentrations add quantitative depth to the qualitative richness of the chemical composition. However, there are some limitations to the proposed method. In the quantitative assessment, a semi-quantitation was conducted, multiple detected substances were quantified based on a standard with a similar, yet different, chemical structure, as detailed above. Hence, a variation in the ionization between a standard and analytes may be a limitation. Future quantification based on the individual isolated secondary metabolites is recommended. In addition, isolation and accurate identification of the newly annotated caffeic acid oligomers and caffeoylhexaric acids will strengthen the validity of the present work.

## 3. Materials and Methods

### 3.1. Chemicals

Acetonitrile (hypergrade for LC–MS), formic acid (for LC–MS), and methanol (analytical grade) were purchased from Chromasolv (Sofia, Bulgaria). The reference standards used for compound identification were obtained from Extrasynthese (Genay, France) for protocatechuic, gentisic acids, and apigenin. Chlorogenic, caffeic, rosmarinic, cichoric acid, pectolinarigenin, and scutellarein were supplied from Phytolab (Vesten-bergsgreuth, Bavaria, Germany).

### 3.2. Plant Material

*S. scardica* seedlings were bought from a certified greenhouse “Mursalski-biogroup” (Bulgaria) and subsequently bred on alluvial soil with sunny exposure in an herbal garden (Rayanovtsi village, Vidin region) in Bulgaria at 349 m a.s.l. (43.7023° N 22.5206° E). The plant was identified by one of the authors (D.Z.) according to Assenov (1989) [31]. The plant material (aerial parts of 4-year-old plants) was collected during the flowering stage in July 2022 and dried for one week in the shade at room temperature. Then it was comminuted with a grinder (Rohnson, R-942, 220–240 V, 50/60 Hz, 200 W, Prague, Czech Republic) and stored in a dry and cool place until further analysis. The fresh/dried mass ratio is 4:1.

### 3.3. Sample Extraction

Air-dried aerial parts (100 g) were infused twice with boiled water (1:20 *w*/*v*) and extracted for 15 min at room temperature. The herbal infusion was lyophilized (lyophilizer Biobase BK-FD10P, BIOBASE, Jinan, China) to yield crude extracts of 12.5 g.

### 3.4. UHPLC–HRMS Dereplication/Annotation

The UHPLC–HRMS analyses were performed, as described previously [22], using a Q Exactive Plus mass spectrometer (Thermo Fisher Scientific, Inc., Waltham, MA, USA) equipped with a heated electrospray ionization (HESI-II) probe (Thermo Scientific). The equipment was operated in negative ion mode within the *m*/*z* range from 130 to 2000 at a resolution of 70,000. Other instrument parameters for full MS mode were set as follows: automatic gain control (AGC) target 3 × 10^6^, maximum injection time (IT) 100 ms, number of scan ranges 1. For the DD-MS^2^ mode, the instrument parameters were as follows: microscans 1, resolution 17,500, AGC target 1 × 10^5^, maximum IT 50 ms, MSX count 1, Top5, isolation window 2.0 *m*/*z*, stepped normalized collision energy (NCE) 10, 20, 60 eV. The chromatographic separation was achieved on a reversed phase column Kromasil EternityXT C18 (1.8 µm, 2.1 × 100 mm) at 40 °C. The UHPLC analyses were run with a mobile phase containing 0.1% formic acid in water (A) and 0.1% formic acid in acetonitrile (B). The run time was 33 min. The gradient elution program was used as follows: 0–1 min, 0–5% B; 1–20 min, 5–30% B; 20–25 min, 30–50% B; 25–30 min, 50–70% B; 30–33 min, 70–95%; 33–34 min 95–5%B. The equilibration time was 4 min. The injection volume and the flow rate were set to 1 µL and 300 µL/min, respectively. Data acquisition was performed using Xcalibur 4.2 (Thermo Scientific, Waltham, MA, USA) instrument control/data handling software.

### 3.5. UHPLC–HRMS Quantification

The UHPLC–HRMS quantification was conducted using the external standard method. Standard calibrations of protocatechuic (**12**), gentisic (**21**), caffeic (**20**), rosmarinic (**27**), chlorogenic (**36**), cichoric acids, apigenin, and scutellarein were established at five data points covering the concentration range of each analyte according to the level expected in the plant samples. Working solutions containing 0.05, 0.025, 0.012, 0.006, and 0.003 mg/mL of the assayed analytes were prepared from a stock solution in methanol containing 0.1 mg/mL. Based on the similar structure, the quantity of **9**, **10**, **14**, and **21** was determined based on the calibration curve of gentisic acid; **11**, **12**, **18**, and **24** as protocatechuic acid; **13**, **14**, **20**, **51**–**56**, and **63** as caffeic acid; **27**–**31** as rosmarinic acid; **36**, **38**, **39**, **41**–**44** as chlorogenic acid; **65**, **66**, **69**, **75**, **77**, **86**–**91** as apigenin; while **70**–**74**, **76**, **78**–**84**, and **88** as scutellarein. Regression equations were as follows: gentisic acid y = 1,977,866,070x + 2460.9184 (R^2^ = 0.9951) protocatechuic acid y = 1,416,556,589.51x + 1581.8936 (R^2^ = 0.9965); caffeic acid y = 1,133,573,161x − 410.2916 (R^2^ = 0.9928); rosmarinic acid y = 2,190,130,610x + 22,248.9144 (R^2^ = 0.9950); chlorogenic acid y = 4,602,799,047.3258x − 921.0750 (R^2^ = 0.9926); cichoric acid y = 4,589,524,443.3458x + 85.8499 (R^2^ = 0.9999); apigenin y = 4,594,027,463.2779x + 21,088.0103 (R^2^ = 0.9983); scutellarein y = 1,976,006,256x − 41,744.9522 (R^2^ = 0.9420). The peak areas were calculated by integrating the Area Under the Curve (AUC) of the full-scan intensity scans for the corresponding molecular ion. These scans were also filtered for the presence of the characteristic base peak. MZmine 2.53 software was applied to the UHPLC–HRMS raw files of the studied *S. scardica* lyophilized infusion to obtain the peak area in the quantitative analysis. Results are expressed as mg/g lyophilized infusion.

## 4. Conclusions

In conclusion, an in-depth phytochemical analysis of *S. scardica* infusion using UHPLC–HRMS was performed. More than 100 metabolites, including sugar acids and saccharides, carboxylic, hydroxybenzoic, hydroxycinnamic, acylquinic and acylhexaric acids, caffeic acids oligomers, phenylpropanoid and iridoid glycosides, flavonoids, fatty acids, and organosulfur compounds were dereplicated/annotated. In addition, 62 metabolites of *S. scardica* were quantified. The presented extraction method of infusion is similar to the approach used in traditional medicine to process *S. scardica* tea. Therefore, the performed state-of-the-art phytochemical analysis of *S. scardica* provide additional knowledge with respect to the chemical constituents of this valuable medicinal plant.

## Figures and Tables

**Figure 1 molecules-29-00204-f001:**
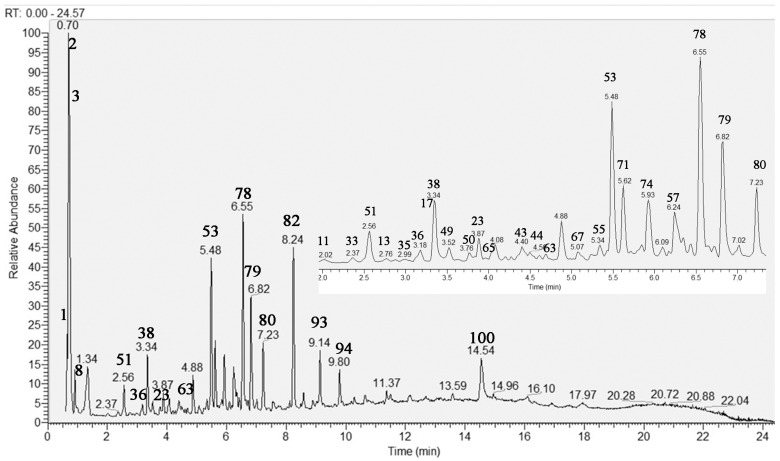
Total ion chromatogram (TIC) in negative ion mode of *Sideritis scardica* extract; the same chromatogram 2–7 min. For compound numbering see Table 1.

**Figure 2 molecules-29-00204-f002:**
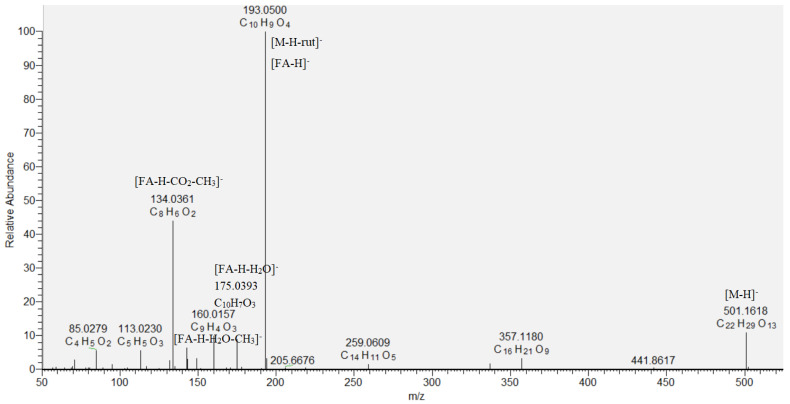
MS/MS spectrum of ferulic acid *O*-rutinoside (**23**).

**Figure 3 molecules-29-00204-f003:**
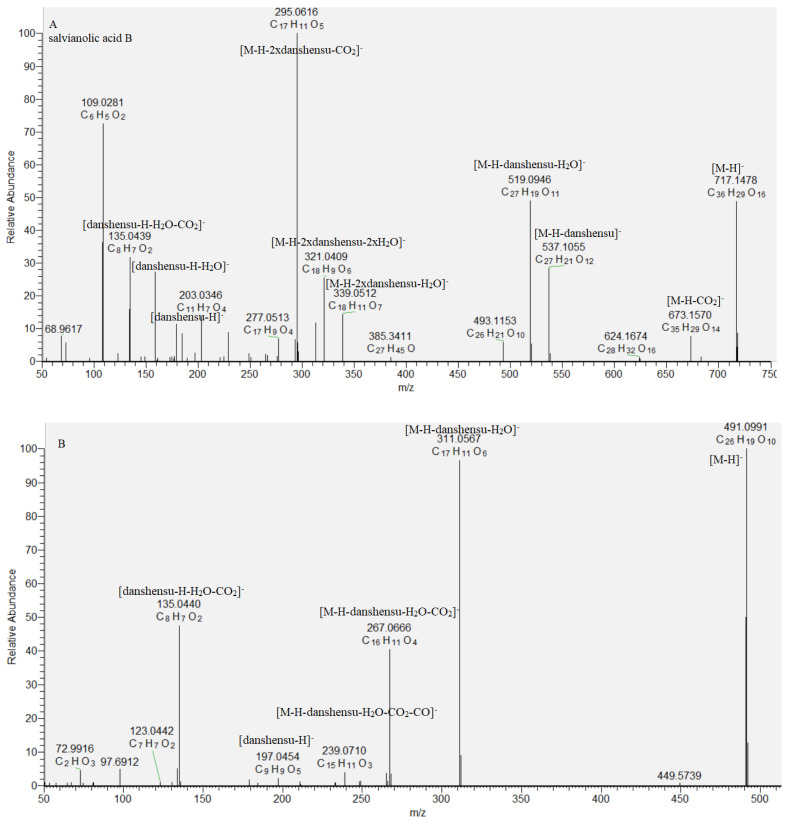
MS/MS spectrum of (**A**) salvianolic acid B (**28**) and (**B**) isosalvianolic acid C (**30**).

**Figure 4 molecules-29-00204-f004:**
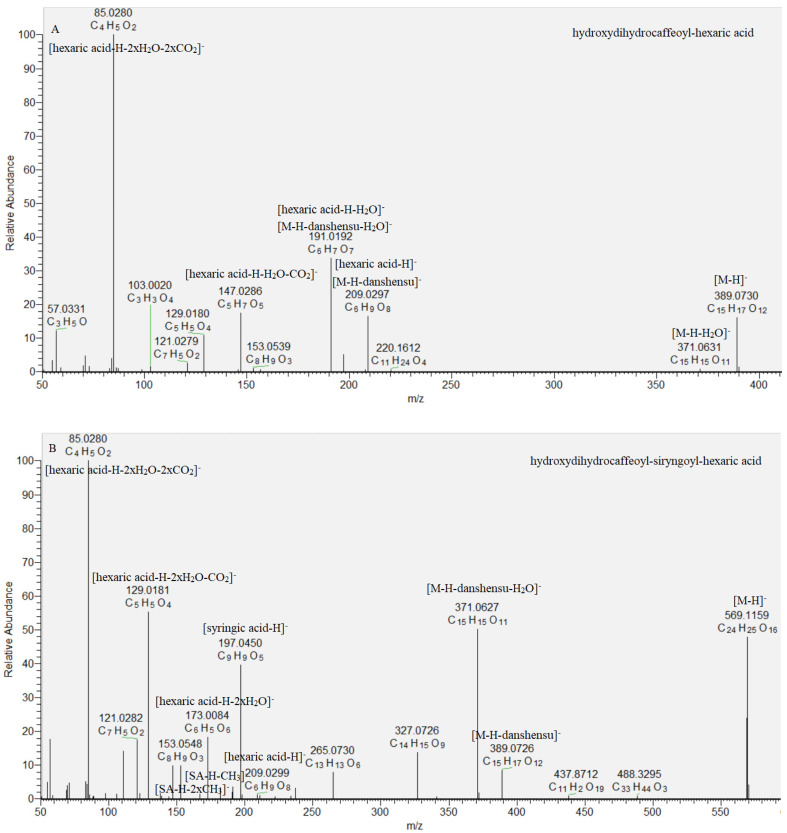
MS/MS spectrum of (**A**) hydroxydihydrocaffeoyl–hexaric acid (**47**) and (**B**) hydroxydihydrocaffeoyl–siryngoyl–hexaric acid (**49**).

**Figure 5 molecules-29-00204-f005:**
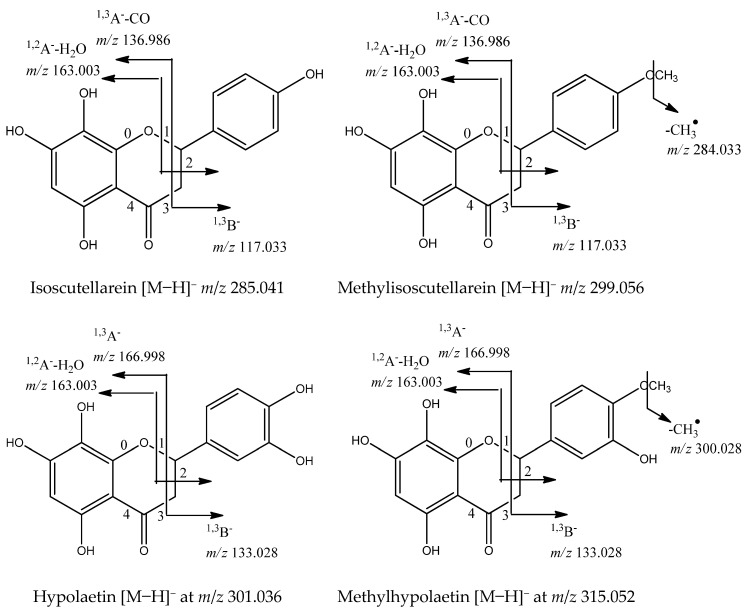
Fragmentation pathways for flavones aglycons caused by cleavage of C-ring bonds in negative ion mode.

## Data Availability

Data are contained within the article.

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
