# Peer review of "A Comprehensive Phytochemical Analysis of Sideritis scardica Infusion Using Orbitrap UHPLC-HRMS"

_molecules, 2023, doi:10.3390/molecules29010204_

Round 1

Reviewer 1 Report

Comments and Suggestions for Authors

The study provides detailed LC-HR(Orbitrap)MS data on phytochemical profile of a Sideritis scardica infusion, which is of interest for this journal readership. The method of analysis is reliable and useful as reference for further studies on this important medicinal and dietary plant. However, there are quite many comments and suggestions to be considered by the Authors before publication:

1. The title is actually quite odd. "State-of-the art" is rather a prerequisite of all scientific  research, so doesn't need to be mentioned in the title! Also, the word "performance" does not bring  about any additional meaning. Performance would make sense if there was a comparison between different HR techniques. An example of a more straightforward title: A comprehensive phytochemical analysis of Sideritis scardica infusion using Orbitrap LC-HR-MS. However, the Authors can of course modify it as they wish;

2. The Introduction is quite full of confusing, general and odd phrasing. For example,:

line 32. why medicinal plants "appear to be a complex matrix ... "and so on?

the sentences in lines 33, 34 are also rather trivial  and not really needed here. The same for the following: "The use of UHPLC suggests (why suggests?) flexible (what do you mean by it?) .... separation.  So,  please, try to rewrite the opening paragraph in a more understandable way, or perhaps just skip it, starting with the plant characterization.

line 41 and on: please, try to avoid non-uniform naming when providing local/verrnacular terms -for example mixing Bulgarian with English (Mursalski tea, while Sharplaninski chai), whereas Greek is just translated. Perhaps also this part is not necessary - there are hundreds of scientific and popular literature sources on this plant.

Further, be careful with pharmacological properties description. for example, ref [8] is a study on rats, so can't be an ultimate evidence that "regular consumption... ca promote weight loss ... etc";

Line 55 - "the plant's beneficial application of the species ..."  sounds odd, please rewrite.

line 60: maybe better not to distinguish  "traditional" and "modern" extractions (SCE is actually old enough). Just list them, please, as "various extraction techniques (....)";

In general, the English of this section needs thorough proofreading and editing.

RESULTS and DISCUSSION

Line 86 and on. I would rather avoid claims that this is a first analysis - it has been studied before using also LC-MS, but not high-res and not to such extent. So, please, rewrite this part to avoid unfounded claims. Your study is an important contribution to phytochemistry, but the cited previous works were also important. It is enough to mention specifically what has been novel.

Figure 1 is difficult to read - please, enlarge the chromatogram parts (e.g minutes 2-7 and insert it s a cropped thumbnail). 

Also, I suggest changing the formula of the Table 1 (and accordingly peak numbering) to: number according to tR. Separate free hydroxycinnamic, benzoic and quinic/oxaloglutaric/citric (non-aromatic).

The quantitative analysis raises some doubts. specifically, the sum of all compounds per freeze-dried infusion exceeds 900 mg/gram. It is quite impossible (no ballast carbohydrates? really?). Please, discuss it. Also, it is unclear how these were measured - from absorbance or from mass spec? Apparently it was mass spec, then ionization efficacy may have influenced it and the method should have been validated for each phytochemical class separately! If no possibility, at least a thorough discussion on limitations of this approach will be mandatory.

Methods

1. Plant material has been obtained from a vendor. So, was it cultivated? If so, the agronomic details should be provided. Also, the genotype/variety/accession/clone or any other biological origin or identification would be necessary. 

2. Fresh/dried  mass ratio should be established. How long were the plants dried. Was the herb comminuted before drying (or after?)

3. What do you mean by "hot water"? How could the herb be soaked in hot water at room temperature?! What were the conditions of freeze-drying? Has this whole method been optimized somehow? 

4. As mentioned above, how the peak areas were determined and how exactly were the calibrations constructed?

Comments on the Quality of English Language

The English requires revision, especially in the non-technical part (Introduction and Discussion)

Author Response

The study provides detailed LC-HR(Orbitrap)MS data on phytochemical profile of a Sideritis scardica infusion, which is of interest for this journal readership. The method of analysis is reliable and useful as reference for further studies on this important medicinal and dietary plant. However, there are quite many comments and suggestions to be considered by the Authors before publication:

  1. The title is actually quite odd. "State-of-the art" is rather a prerequisite of all scientific  research, so doesn't need to be mentioned in the title! Also, the word "performance" does not bring  about any additional meaning. Performance would make sense if there was a comparison between different HR techniques. An example of a more straightforward title: A comprehensive phytochemical analysis of Sideritis scardica infusion using Orbitrap LC-HR-MS. However, the Authors can of course modify it as they wish;

Response: Dear Reviewer, Thanks for the recommendation. The title was changed as follows: A Comprehensive Phytochemical Analysis of Sideritis scardica Infusion using Orbitrap UHPLC-HRMS (See page 1).

  1. The Introduction is quite full of confusing, general and odd phrasing. For example,:

line 32. why medicinal plants "appear to be a complex matrix ... "and so on?

the sentences in lines 33, 34 are also rather trivial  and not really needed here. The same for the following: "The use of UHPLC suggests (why suggests?) flexible (what do you mean by it?) .... separation.  So,  please, try to rewrite the opening paragraph in a more understandable way, or perhaps just skip it, starting with the plant characterization.

line 41 and on: please, try to avoid non-uniform naming when providing local/verrnacular terms -for example mixing Bulgarian with English (Mursalski tea, while Sharplaninski chai), whereas Greek is just translated. Perhaps also this part is not necessary - there are hundreds of scientific and popular literature sources on this plant.

Further, be careful with pharmacological properties description. for example, ref [8] is a study on rats, so can't be an ultimate evidence that "regular consumption... ca promote weight loss ... etc";

Line 55 - "the plant's beneficial application of the species ..."  sounds odd, please rewrite.

line 60: maybe better not to distinguish  "traditional" and "modern" extractions (SCE is actually old enough). Just list them, please, as "various extraction techniques (....)";

In general, the English of this section needs thorough proofreading and editing.

Response: Thanks for the comments. The introduction was rewritten according to the Reviewer’s recommendation (See page 1, Introduction).

RESULTS and DISCUSSION

Line 86 and on. I would rather avoid claims that this is a first analysis - it has been studied before using also LC-MS, but not high-res and not to such extent. So, please, rewrite this part to avoid unfounded claims. Your study is an important contribution to phytochemistry, but the cited previous works were also important. It is enough to mention specifically what has been novel.

Response: Thanks for the comments. All claims for the first LC-MS analysis were removed.

Figure 1 is difficult to read - please, enlarge the chromatogram parts (e.g minutes 2-7 and insert it s a cropped thumbnail). 

Response: Figure 1 was changed according to the reviewer’s recommendation.

Also, I suggest changing the formula of the Table 1 (and accordingly peak numbering) to: number according to tR. Separate free hydroxycinnamic, benzoic and quinic/oxaloglutaric/citric (non-aromatic).

Response: Dear reviewer, thanks for the suggestion. We arranged compounds in different group of secondary metabolites and according to their tR. Quinic, oxaloglutaric and citric acids were separate in a group of carboxylic acids according to your recommendation.

The quantitative analysis raises some doubts. specifically, the sum of all compounds per freeze-dried infusion exceeds 900 mg/gram. It is quite impossible (no ballast carbohydrates? really?). Please, discuss it. Also, it is unclear how these were measured - from absorbance or from mass spec? Apparently it was mass spec, then ionization efficacy may have influenced it and the method should have been validated for each phytochemical class separately! If no possibility, at least a thorough discussion on limitations of this approach will be mandatory.

Response: As a limitation of the quantitative assessment, a semi-quantitation was conducted, multiple detected substances were quantified based on a standard with similar, but still different chemical structure, as detailed above. Hence, a variation in the ionization between a standard and analytes, may be a contributing limitation. Moreover, the presented quantity of compounds is for the lyophilized infusion of S. scardica leaves, stems and flowers with water, not for extract, or for any organic solvent. In addition, the quantity of all compounds are around 10 % of  the dried plant material. Previously it was found that the content of phenylethanoid glycosides ranged from 1.22 mg/g dry herb for S. lanata to 108.3 mg/g dry herb for S. scardica from Rhodopi Mountain, Bulgaria (Stanoeva et al., 2015).

Methods

  1. Plant material has been obtained from a vendor. So, was it cultivated? If so, the agronomic details should be provided. Also, the genotype/variety/accession/clone or any other biological origin or identification would be necessary. 

Response: Dear Reviewer, indeed, we realize that the original version of this section was missing details. The seedlings were bought from a farmer from a certificated greenhouse Mursalski-biogroup” (Bulgaria) (https://mursalski-biogroup.com/). Then the seedlings were bred in a private herbal garden according to the agronomic recommendation: no special preference to soil type, sunny exposure of the field/garden, regular feeding of the soil with nutrients as nitrogen (N), phosphorus (P) and potassium (K), and etc.

The Plant material section is rewritten according to the comments. (See page Materials and methods, Plant material).

  1. Fresh/dried  mass ratio should be established. How long were the plants dried. Was the herb comminuted before drying (or after?)

Response: The plant material (aerial parts of 4-year-old plants) was collected during full flowering stage in July 2022 and dried for one week under shade at room temperature. Then it was comminuted with grinder (Rohnson, R-942, 220-240V, 50/60Hz, 200W) and stored in a dry and cool place until used for further analysis. Fresh/dried mass ratio is 4:1. (See page Materials and methods, Plant material).

  1. What do you mean by "hot water"? How could the herb be soaked in hot water at room temperatu re?! What were the conditions of freeze-drying? Has this whole method been optimized somehow? 

Dear Reviewer, thank you for the questions so as to clarify the extraction procedure. The goal was to make an herbal infusion like people usually do at home. The procedure was done as follow: The plant material was infused with boiled water (100oC). Then the herbal infusion was lyophilized at the following conditions: -60 ˚, 72 h.

The text is rewritten in accordance with the comments (See Materials and methods, Sample extraction).

  1. As mentioned above, how the peak areas were determined and how exactly were the calibrations constructed?

Response: The peak areas were calculated by integrating the Area Under the Curve (AUC) of the full-scan intensity scans for the corresponding molecular ion. These scans were also filtered for the presence of the characteristic base peak.

Comments on the Quality of English Language

The English requires revision, especially in the non-technical part (Introduction and Discussion

Response: Done

Reviewer 2 Report

Comments and Suggestions for Authors

Brief summary:

The manuscript deals with the phytochemical identification of compounds in Sideretis scardica lyophilized herbal infusion by liquid chromatography hyphenated to high-resolution mass spectrometry using dereplication. A lot of compounds were subsequently quantified.

General comments:

The title of the article sounds a little bit pompous, since phytochemical identification seems to have been performed manually in accordance with previous studies and did not use bioinformatic tools like molecular networks to group compounds by spectral similarities or other annotation tools to look into databases. Something like “Phytochemical Analysis of Sideritis scardica by UHPLC-HRMS Orbitrap” seems more realistic.

L. 101: compound 2 was identified as glucose. It could be more prudent to express the identification with the generic term “hexose”.

L. 128: the trimer for caffeic acid is mentioned as compound 30, but the formula of compound 28 looks to better fit with the definition of the beginning of the paragraph. L. 130: rosmarinic acid is compound 27, not 13.

Several times the authors use the term “base peak” but it’s not necessary the case: the base peak is the most intense ion of a spectrum (and often the representation of the base peak chromatogram (BPC) looks better than the one of the total ion chromatogram (TIC)). For example, L. 142, L. 161 (compound 40), L. 182.

Several times, the authors use recent publications to justify the dereplication of compounds: these publications do not necessary give the proof of spectral similarities, but refer to other publications (which can do the same). If possible, the authors should cite the original paper, or the comparison with a standard (compounds with a 1 as level of confidence) is a more convincing proof. Example L. 155 with reference 16.

L. 163-170: the authors have to carefully use the data published in reference 21 to compare with their own data since it’s not the same mass spectrometer geometry and especially the fragmentation mode: ion trap for LCQ Deca XP versus collision cell for the Q-Exactive. So the observed ratio between ions are not necessarily transposable.

L. 197: m/z 179.03 is not a neutral loss of caffeic acid, but a fragment ion.

L. 222-223: compound 88 appears two times as MIS and MHL

L. 239: compound 99 is not monounsaturated but polyunsaturated (C18:2), but compounds 94 and 97 are.

L. 231-232 and Figure 5: the authors have to check the fragmentation pathways since the one presented are not coherent:

- isoscutellarein: 1,2A- -CO is not equal to 136.986 (maybe 1,3A- -CO ?)

- m/z of 1,3B- for the compounds of the left could not be equals to the ones of the right where there is a supplementary methoxy

- methylhypolaetin: 1,2A- -CO is not equal to 136.986

Note: normally the arrows should be oriented in the direction of the ion (globally the x,yA arrows should point to the left).

The authors could merge Table 2 with Table 1, simply by adding a column.

L.310-323: the authors said they use the mass spectrometer in negative and positive ion mode, but in this publication, I see only results in negative one. In this part, the authors should add some information about the setting of the mass spectrometer like (at least) the resolution used to perform the acquisitions in MS and MS/MS, the energy of collision MS/MS experiments. Rem: the flow rate is mentioned twice. L. 317 “the run time” -> “the gradient time” since the run is longer.

Specific comments:

L. 24-26: mg/g di -> mg/g li ?

L. 104: Da) units

L. 113: phenolic acid glycosides / neutral losses

Figures 3 and 4 with MS/MS spectra: the authors could add the name of the compound directly on the spectrum for a better readability.

L. 190, 229: after a loss of CH3, the anion become a radical anion: [Agl-H-CH3]-

L. 203: loss of glucose -> losses of hexose and…

L. 257: acidified with formic acid

L. 297: acidsq

L. 316: contained -> containing

Author Response

Brief summary:

The manuscript deals with the phytochemical identification of compounds in Sideretis scardica lyophilized herbal infusion by liquid chromatography hyphenated to high-resolution mass spectrometry using dereplication. A lot of compounds were subsequently quantified.

General comments:

The title of the article sounds a little bit pompous, since phytochemical identification seems to have been performed manually in accordance with previous studies and did not use bioinformatic tools like molecular networks to group compounds by spectral similarities or other annotation tools to look into databases. Something like “Phytochemical Analysis of Sideritis scardica by UHPLC-HRMS Orbitrap” seems more realistic.

Response: Dear Reviewer, thanks for the recommendation. The title was changed as follows: A Comprehensive Phytochemical Analysis of Sideritis scardica Infusion using Orbitrap UHPLC-HRMS (See page 1).

L. 101: compound 2 was identified as glucose. It could be more prudent to express the identification with the generic term “hexose”.

Response: Thanks for the comment. The name was changed to hexose (See Results and discussion, row 102 and Table 1).

L. 128: the trimer for caffeic acid is mentioned as compound 30, but the formula of compound 28 looks to better fit with the definition of the beginning of the paragraph. L. 130: rosmarinic acid is compound 27, not 13.

Response: Thanks for the useful comment. The numbers of compound were rearranged and changed. You are absolutely right. Rosmarinic acid is number 29. There are two trimers lithospermic acid (27) and isosalvianolic acid C (30), and two tetramers salvianolic acid B (28) and didihydrosalvianolic acid B (31). See 2.1.3. Caffeic acids oligomers.

Several times the authors use the term “base peak” but it’s not necessary the case: the base peak is the most intense ion of a spectrum (and often the representation of the base peak chromatogram (BPC) looks better than the one of the total ion chromatogram (TIC)). For example, L. 142, L. 161 (compound 40), L. 182.

Response: Thanks for the comment. The use of the term “base peak” is absolutely necessary in these cases, because it is a part of the fragmentation pathway of the compounds, with regards to their MS/MS spectra. In the MS/MS spectrum of acylquinic acids there are a base peak, and fragment ions, indicating the position of the acyl residue in the quinic acid skeleton. For example, a base peak at m/z 173 in the MS/MS spectra of these compounds, indicated a presence of acyl residue in the 4-position of the quinic acid, while a base peak at m/z 191,055 is indicative for the position 5. TIC was presented only in Figure 1. All other Figures and discussion concerned MS/MS spectra.

Several times, the authors use recent publications to justify the dereplication of compounds: these publications do not necessary give the proof of spectral similarities, but refer to other publications (which can do the same). If possible, the authors should cite the original paper, or the comparison with a standard (compounds with a 1 as level of confidence) is a more convincing proof. Example L. 155 with reference 16.

Response: Thanks for the comment. The reference 16.Gevrenova, R.; Zengin, G.; Sinan, K. I.; Zheleva-Dimitrova, D.; Balabanova, V.; Kolmayer, M.; Voynikov, Y.; Joubert, O., An In-Depth Study of Metabolite Profile and Biological Potential of Tanacetum balsamita L. (Costmary). Plants 2023, 12, (1), 22. is a part of our long experience with acylquinic acids dereplication in many Asteraceae species. Based on a comparison with different reference standards and based on Clliford`s hierarchical key we created an in-house ORBITRAP strategy for acylquinic acid` dereplication. A new reference was included in the text (See Results and Discussion 2.1.4. Acylquinic acid)

L. 63-170: the authors have to carefully use the data published in reference 21 to compare with their own data since it’s not the same mass spectrometer geometry and especially the fragmentation mode: ion trap for LCQ Deca XP versuscollision cell for the Q-Exactive. So the observed ratio between ions are not necessarily transposable.

Response: Thanks for the comment. New references were included (See Results and Discussion 2.1.4. Acylquinic acid)

L. 197: m/z179.03 is not a neutral loss of caffeic acid, but a fragment ion.

Response: Thanks for the comment. There is no the term “neutral” in the line 197, but the sentence was precise (See Results and Discussion).

L. 222-223: compound 88 appears two times as MIS and MHL

Response: Thanks for the comment. The sentence was corrected (See Results and Discussion 2.1.8. Flavonoids).

L. 239: compound 99 is not monounsaturated but polyunsaturated (C18:2), but compounds 94 and 97 are.

Response: Thanks for the comment. The sentence was corrected (See Results and Discussion 2.1.9. Fatty acids and organosulfur compounds).

L. 231-232 and Figure 5: the authors have to check the fragmentation pathways since the one presented are not coherent:

- isoscutellarein: 1,2A- -CO is not equal to 136.986 (maybe 1,3A- -CO ?)

m/z of 1,3B- for the compounds of the left could not be equals to the ones of the right where there is a supplementary methoxy

- methylhypolaetin: 1,2A- -CO is not equal to 136.986

Note: normally the arrows should be oriented in the direction of the ion (globally the x,yA arrows should point to the left).

Response: Thanks for the valuable comments. The figure was changed according to the reviewer’s recommendation. In our conditions, reference standards with 4'-methoxy group (pectolinariganen for example) gave a fragment ion at m/z 117, corresponding to 1,3B--CH3. In addition, most of methoxylated derivatives of isoscutellarein and hypolaetin, previously found in S. scardica have OCH3 groups in position 4' (Żyżelewicz et al., 2020; Petreska et al., 2011).

The authors could merge Table 2 with Table 1, simply by adding a column.

Response: Thanks for the comment. Tables were merged.

L.310-323: the authors said they use the mass spectrometer in negative and positive ion mode, but in this publication, I see only results in negative one. In this part, the authors should add some information about the setting of the mass spectrometer like (at least) the resolution used to perform the acquisitions in MS and MS/MS, the energy of collision MS/MS experiments. Rem: the flow rate is mentioned twice. L. 317 “the run time” -> “the gradient time” since the run is longer.

Response: Thanks for the valuable comments. The negative ion mode is more informative for all of compounds, moreover some groups like hexaric and fatty acids can be seen only in negative ion mode. The sentence was precise. The conditions for the MS/MS experiments were added.

Specific comments:

L. 24-26: mg/g di -> mg/g li ?

Response: Thanks for the comment. The sentence was corrected

L. 104: Da) units

Response: Thanks for the comment. The sentence was corrected

L. 113: phenolic acid glycosides / neutral losses

Response: Thanks for the comment. The sentence was corrected

Figures 3 and 4 with MS/MS spectra: the authors could add the name of the compound directly on the spectrum for a better readability.

Response: Thanks for the comment. The figures were changed.

L. 190, 229: after a loss of CH3, the anion become a radical anion: [Agl-H-CH3]-

Response: Thanks for the comment. The sentences were corrected

L. 203: loss of glucose -> losses of hexose and…

Response: Thanks for the comment. The sentence was corrected

L. 257: acidified with formic acid

Response: Thanks for the comment. The sentence was corrected

L. 297: acidsq

Response: Thanks for the comment. The sentence was corrected

L. 316: contained -> containing

Response: Thanks for the comment. The sentence was corrected

Reviewer 3 Report

Comments and Suggestions for Authors

This manuscript presents a state-of-the-art phytochemical analysis of Sideritis scardica, commonly known as "mountain tea" or "Olympus tea," an endemic plant from the Balkan Peninsula. The use of ultra-high-performance liquid chromatography hyphenated with high-resolution mass spectrometry (UHPLC-HRMS) for the in-depth analysis is commendable and adds valuable insights into the chemical composition of this medicinal plant.

The study employed the external standard method for quantitative determination of the main secondary metabolites, revealing a rich diversity of over 100 metabolites in S. scardica infusion.

A notable contribution of this study is the first-time dereplication and fragmentation patterns of 5 caffeic acids oligomers and 4 acylhexaric acids in S. scardica, expanding the current understanding of its chemical profile.

The quantitative analysis identified major compounds in S. scardica infusion, with phenylethanoid verbascoside, glycosides of isoscutellarein, methylisoscutelarein, hypolaetin, and caffeic acid standing out as significant constituents. The reported concentrations add quantitative depth to the qualitative richness of the chemical composition.

Overall, this state-of-the-art phytochemical analysis significantly contributes to our understanding of the chemical constituents of S. scardica, a valuable medicinal plant. The findings not only expand the knowledge base but also have implications for the potential therapeutic uses of this plant. The manuscript is well-structured, and the methodology employed is robust, making it a valuable addition to the field of phytochemical research.

It is suggested to include three headings 1) Study strength 2)Study limitation 3)Future direction

It is suggested to explain your discussion with more indepth, including last five year reference.

Author Response

This manuscript presents a state-of-the-art phytochemical analysis of Sideritis scardica, commonly known as "mountain tea" or "Olympus tea," an endemic plant from the Balkan Peninsula. The use of ultra-high-performance liquid chromatography hyphenated with high-resolution mass spectrometry (UHPLC-HRMS) for the in-depth analysis is commendable and adds valuable insights into the chemical composition of this medicinal plant.

 The study employed the external standard method for quantitative determination of the main secondary metabolites, revealing a rich diversity of over 100 metabolites in S. scardica infusion.

 A notable contribution of this study is the first-time dereplication and fragmentation patterns of 5 caffeic acids oligomers and 4 acylhexaric acids in S. scardica, expanding the current understanding of its chemical profile.

The quantitative analysis identified major compounds in S. scardica infusion, with phenylethanoid verbascoside, glycosides of isoscutellarein, methylisoscutelarein, hypolaetin, and caffeic acid standing out as significant constituents. The reported concentrations add quantitative depth to the qualitative richness of the chemical composition.

Response: Dear reviewer, thanks for the comments.

Overall, this state-of-the-art phytochemical analysis significantly contributes to our understanding of the chemical constituents of S. scardica, a valuable medicinal plant. The findings not only expand the knowledge base but also have implications for the potential therapeutic uses of this plant. The manuscript is well-structured, and the methodology employed is robust, making it a valuable addition to the field of phytochemical research.

Response: Dear reviewer, thanks for the comments.

It is suggested to include three headings 1) Study strength 2)Study limitation 3)Future direction

Response: Dear reviewer, thanks for the comments. A new heading, combining 1) Study strength 2)Study limitation 3)Future direction was added (See Results and discussion.

It is suggested to explain your discussion with more indepth, including last five year reference.

Response: Dear reviewer, thanks for the comments. The discussion was improved.